# Exploring the Influence of Age, Gender and Body Mass Index on Colorectal Cancer Location

**DOI:** 10.3390/medicina59081399

**Published:** 2023-07-30

**Authors:** Dorel Popovici, Cristian Stanisav, Sorin Saftescu, Serban Negru, Radu Dragomir, Daniel Ciurescu, Razvan Diaconescu

**Affiliations:** 1Department of Oncology, Faculty of Medicine, Victor Babeş University of Medicine and Pharmacy Timisoara, Eftimie Murgu Square 2, 300041 Timisoara, Romania; 2Departments of Radiology, Victor Babeş University of Medicine and Pharmacy Timisoara, 300041 Timisoara, Romania; 3Departments of Obstetrics and Gynecology, Victor Babeş University of Medicine and Pharmacy Timisoara, 300041 Timisoara, Romania; 4Departments of Medical and Surgical Specialties, Faculty of Medicine, Transylvania University of Brașov, 500019 Brasov, Romania; 5Departments of General Surgery, Vasile Goldiş Western University of Arad, 310025 Arad, Romania

**Keywords:** colorectal cancer, rectal cancer, obesity, body mass index (BMI), patient demographics, adenoma detection, SIRT6 loss, adipose tissue homeostasis, visceral obesity, gut microbiome, systemic gene expression

## Abstract

*Background and objectives*: The global burden of non-communicable diseases like obesity and cancer, particularly colorectal cancer (CRC), is increasing. The present study aimed to investigate the association between CRC location (proximal vs. distal) and patient demographic factors including age, sex, and BMI, as well as cancer stage at diagnosis. *Materials and Methods*: In this cross-sectional study, data from 830 patients diagnosed with CRC were analyzed. The variables included age, sex, weight, height, BMI, cancer location, and cancer stage at diagnosis. Patients were stratified into three age groups and three BMI categories, and we analyzed the association between cancer location and these variables using Chi-squared tests and multivariate logistic regression. *Results*: The rectum and ascending colon were the most common locations of malignant neoplasms. No statistically significant differences in cancer location across age groups were observed. Significant differences were found in the BMI across age groups, particularly in the normal weight and overweight categories. Normal weight and obese patients had a higher proportion of Stage 3 and Stage 4 cancers. Obesity emerged as a significant predictor for rectal cancer in a multivariate logistic regression analysis, with an odds ratio of 1.56. However, no significant associations were found between cancer location and other factors like age, gender, or cancer stage. *Conclusions*: Our study revealed that normal weight and obese patients had a higher proportion of Stage 3 and Stage 4 cancers, with obesity emerging as a significant predictor for rectal cancer. It is important to note that while obesity was found to be a significant predictor for rectal cancer, the development and location of colorectal cancer is likely influenced by various factors beyond those studied here. Therefore, further research is needed to investigate the roles of other potential risk factors, like loss of SIRT6 and adipose tissue homeostasis. Additionally, inflammation associated with microbiota in the colorectal mucosa, systemic gene expression, and visceral obesity may also play important roles in the development and progression of colorectal cancer. Understanding these intricate relationships is crucial for better screening, disease prognosis, and management strategies.

## 1. Introduction

The global burden of chronic diseases such as obesity and cancer has been escalating over the years. These conditions not only affect the quality of life but also pose a significant challenge to healthcare systems worldwide. Obesity has been labeled as a worldwide health burden, with an increasing trend observed in the percentage of deaths attributed to this condition, rising from 4.5% in 1990 to 8% in 2017. The concern regarding obesity is especially pronounced in more affluent countries across Europe, North America, and Oceania, where over 15% of deaths were linked to this health issue in 2017. Despite the lower obesity rates in South Asia and Sub-Saharan Africa, the adverse effects of obesity should not be ignored [1].

Cancer is another non-communicable disease causing significant mortality and morbidity globally. According to GLOBOCAN data, in 2020, the worldwide burden of cancer escalated to 19.3 million new cases and 10 million deaths, and the five-year prevalence of people living after a cancer diagnosis was approximated to be 50.6 million. The most common cancer types include breast, lung, and colorectal cancers, with their prevalence varying between genders [2].

The importance of early cancer detection cannot be overstated. Earlier diagnosis leads to better survival rates, improved patient experiences, a lower treatment morbidity, and a superior quality of life. Two patient behaviors that can aid in early cancer diagnosis include attending cancer screenings and presenting promptly to the GPs office if they have any cancer-related symptoms [3].

Colorectal cancer (CRC) is a significant subtype of cancer, where most cases start as pre-cancerous lesions or adenomatous polyps. Although the majority of CRC cases occur in individuals with no family history or apparent genetic predisposition, recent studies have highlighted that about 45–47% of colorectal cancer cases are attributed to modifiable risk factors, primarily alcohol consumption, smoking, processed meat intake, and obesity [4,5,6,7]. 

In a comprehensive population-based cohort study, researchers found that the correlation between excessive weight and the risk of CRC might be considerably stronger than initially thought. While initial analyses suggested little to no link, this association appeared to intensify with a longer follow-up period. The study indicates that long-term obesity could play a critical role in the development of CRC. Remarkably, the body mass index, especially a higher one in the elderly population, was not only associated with an increased incidence of CRC but also with a higher mortality rate from this disease. Furthermore, research suggests that the proportion of CRC cases attributable to overweight and obesity could be higher than earlier estimations. Intriguingly, the links between overweight and obesity and CRC incidence were less pronounced in women compared to men. This discrepancy could potentially be explained by factors such as hormone replacement therapy that post-menopausal women undergo [5,6].

Interestingly, the microbiota, our gut’s resident microbial community, plays a significant role in our health, including immunity regulation, digestion, and nutrition. Dysbiosis, or imbalance in the gut microbiota, has been associated with several pathologies, including CRC. Thus, the microbiota can be a potential therapeutic target for CRC and other diseases, such as obesity and even extra-intestinal cancers [7]. 

Screening is an essential strategy for early detection of CRC. There are several tests available for colorectal cancer screening, including stool-based tests and visual exams, each with its advantages and disadvantages. From non-invasive tests like the Fecal Occult Blood Test (FOBT) and Stool DNA Test, to more invasive techniques like sigmoidoscopy and colonoscopy, each test has its utility and limitations. The choice of screening method often depends on the individual patient’s circumstances and the healthcare provider’s recommendation [4].

In conclusion, obesity and cancer, specifically colorectal cancer, pose a significant health burden globally. The role of early detection through screening, understanding risk factors, like age and BMI, and exploring novel therapeutic targets, such as the gut microbiota, are paramount in mitigating this health crisis. Further research and public health interventions are needed to curb the increasing trends of obesity and cancer worldwide.

The present study hypothesizes that there is a significant association between colorectal cancer location (proximal vs. distal) and patient demographic factors such as age, sex, and BMI, as well as the cancer stage at initial diagnosis. This hypothesis is grounded in the proposition that individual patient characteristics can influence the location of colorectal cancer manifestation, which may in turn have implications for disease prognosis and management.

## 2. Materials and Methods

### 2.1. Criteria

The patient selection for this cross-sectional study was drawn from the Electronic Medical Records at OncoHelp Association Timisoara, after it was approved by the Institutional Review Board. The chosen period for consideration extended from 1 January 2016 to 1 March 2023, focusing on patient age, gender, weight, height, cancer location, and cancer stage. The initial selection comprised 1043 patients. However, the final dataset was refined to exclude certain categories of patients, so as to ensure that the focus remained on the primary research hypothesis. Firstly, 130 patients were excluded from the study if they had received a diagnosis of any other type of neoplasia either in the past or at the time of their colorectal cancer diagnosis. This was essential to limit the potential confounding variables that may influence the outcomes of this study. Secondly, 23 patients were excluded due to missing or incomplete data on TNM staging or cancer stage. Accurate staging information is vital to understand the extent of colorectal cancer in patients, hence the exclusion. The third exclusion criterion was related to body mass index (BMI). If a patient had an underweight BMI, they were excluded from the study, resulting in 26 patients being left out. Finally, the absence of vital information like height or weight led to the exclusion of an additional 33 patients (Figure 1).

After these considerations, a total of 830 patients were found to be eligible for analysis in this study. The BMI of the patients was calculated using the weight and height information documented in the patient’s medical history. However, it is crucial to note that the analysis was based on self-reported height and weight. The potential underestimation of these parameters by the patients could lead to possible deviations in the calculation of BMI, which might affect the interpretation of the study results. As such, the potential for these deviations was considered when analyzing the results of this study.

### 2.2. Statistical Analysis

Patient data were extracted for variables including age, sex, weight, height, BMI, cancer location, and cancer stage at initial diagnosis. Data analysis was performed using EasyMedStat (version 3.24; www.easymedstat.com, accessed on 20 April 2023). For statistical analysis, numeric variables were expressed as means (±SD) and discrete outcomes as absolute and relative (%) frequencies. Patients were divided into three groups based on age. Group comparability was assessed by comparing baseline demographic data and follow-up duration between groups. The normality of continuous data was verified using the Shapiro–Wilk test and heteroskedasticity was tested with Levene’s test. Continuous outcomes were compared using ANOVA, Welch ANOVA, or Kruskal–Wallis tests based on the data distribution. Discrete outcomes were compared using the Chi-squared or Fisher’s exact test as appropriate. The alpha risk was set to 5% and two-tailed tests were applied. The association between cancer location and age was examined using the Chi-squared test. A multivariate logistic regression was conducted to assess the relationship between proximal and distal cancer localization and explanatory variables, including age, gender, BMI, and cancer stage. The data were checked for multicollinearity using, respectively, the White test and the Shapiro–Wilk test. A *p*-value of 0.05 was considered statistically significant.

## 3. Results

The present study involved a comprehensive analysis of 830 patients diagnosed with colorectal cancer, stratified into three distinct age groups: young adults (27–49 years), middle-aged adults (50–74 years), and elderly adults (75–88 years).

### 3.1. Distribution of Cancer Location

An in-depth analysis of the distribution of cancer locations revealed a variable pattern within these age groups. For young adults, the most frequently diagnosed locations were the rectum (13.56%) and ascending colon (12.5%). Middle-aged adults also predominantly presented with cancers located in the rectum (74.86%) and ascending colon (70.83%). In the elderly adult category, the rectum (11.58%) and ascending colon (16.67%) remained the most common locations for malignant neoplasms. Due to their classification of proximal or distal colorectal cancer in the electronic medical record, twenty-two patients were excluded from the table. A statistical analysis of this distribution concluded that the variations in cancer location across the different age groups were not statistically significant, with a *p*-value of 0.502 (Table 1). Twenty-two patients were not included in this table due to unspecified colon cancer location (C18.9: malignant neoplasm: colon, unspecified).

When considering the patient population irrespective of age, the rectum (43.81%) and ascending colon (14.85%) emerged as the most common locations for malignant neoplasms. These were followed by the rectosigmoid junction (9.41%), transverse colon (9.16%), hepatic flexure (7.05%), splenic flexure (6.06%), descending colon (5.94%), and cecum (3.71%) (Figure 2).

### 3.2. Body Mass Index Analysis

The study also analyzed the BMI across the three age groups. Patients were categorized based on their BMI into three groups: normal weight (NW: 18.5–24.9), overweight (OW: 25–29.9), and obese (OB: 30+). All obesity subgroups were combined into one larger group.

Significant differences were found in the BMI across the age groups, specifically in the normal weight (*p* = 0.022) and overweight (*p* = 0.015) categories. However, the obese category showed no statistically significant difference across the age groups (*p* = 0.16) (Table 2, Figure 3).

### 3.3. BMI and Cancer Stages

We further investigated the correlation between BMI categories and different stages of colorectal cancer (Stage 1 to Stage 4). A statistically significant association emerged between the BMI category and cancer stages for Stage 3 (*p* = 0.02) and Stage 4 (*p* = 0.002). In particular, Stage 3 cancer showed a higher proportion of obese patients, while Stage 4 cancer showed a higher proportion of normal weight patients (Table 3).

### 3.4. Multivariate Logistic Regression Analysis

A multivariate logistic regression model was employed to investigate the relationship between the cancer location (rectum and ascending colon being the most common) and various explanatory variables, including age, gender, BMI, and cancer stage.

In the context of BMI, obesity emerged as a significant predictor for rectal cancer, with an odds ratio of 1.56 (*p* = 0.0213), indicating that obese patients had a higher chance of developing rectal cancer compared to those of normal weight or overweight. However, BMI was not found to be a significant predictor for cancer in the ascending colon (Table 4).

For other variables, no significant associations were found between cancer location and age, gender, or cancer stage. These results imply that while obesity might play a role in rectal cancer, the location of colorectal cancer is likely influenced by a myriad of factors beyond those included in this study. Further research is required to elucidate these relationships.

## 4. Discussion

Our study found a significant association between obesity and the risk of CRC. Obese patients (BMI 30.0–47.0) have a higher prevalence of rectal cancer compared to normal-weight patients (18.5–24.9), with an odds ratio of 1.56 (*p* = 0.0213). Our result is in line with the findings of other articles found in the literature, who reported similar associations between obesity and CRC [8,9,10]. A study by Pang et al. revealed that central adiposity, measured by waist circumference (WC) and waist-to-hip ratio (WHR), has a positive association with CRC risk. When BMI and WC were considered together in the same model, the risk estimates associated with WC maintained some level of significance, while those for BMI became non-significant. Other measures, including hip circumference, percent body fat, height adjusted weight, height, weight-to-height ratio, and weight change since the age of 25, were also positively associated with CRC risk, further emphasizing the role of various adiposity measures in assessing the CRC risk [11].

Additionally, our results showed statistically significant differences between age groups in the normal weight and overweight categories. Specifically, we observed an increasing trend in mean weight values as age increases from young adults to elderly adults in the normal weight category (*p* = 0.022), and a similar pattern in the overweight category (*p* = 0.015). However, there was no statistically significant difference between the age groups in the obese category (*p* = 0.16). This suggests that there may be age-related factors that influence weight gain or weight distribution, particularly for those in the normal weight and overweight categories. 

A shift towards a sedentary lifestyle and increased food intake leads to an accumulation of excess visceral fat, adipocyte hypertrophy in subcutaneous fat, and a loss of muscle mass and strength. Larger-sized adipocytes are linked to a higher release of free fatty acids, which can cause insulin resistance and other metabolic disorders. Moreover, excess visceral fat leads to an overabundance of free fatty acids delivered to the liver, leading in time to liver injury and production of pro-inflammatory cytokines, chemokines, and damage-associated molecular patterns [12,13]. As individuals advance into extreme old age, there is a reduction in anabolic hormones, which, coupled with the direct effects of aging and a decline in activity, results in more pronounced muscle atrophy and greater adipose tissue dysfunction. Aging mechanisms operate at both cellular and organismal levels [12].

Further insight into this trajectory was provided by a large-scale cross-sectional study. This study found that major weight gain in adulthood, specifically an increase of 10 kg or more, was associated with unfavorable lifestyle factors, reflecting prolonged maintenance of poor habits. Interestingly, men who were relatively slim as young adults, indicated by a low BMI at age 20, often showed a greater accumulation of visceral fat related to weight gain. These findings underscore the importance of addressing lifestyle factors to prevent major weight gain in adulthood and mitigate its associated health risks. Thus, the transition from young adulthood to old age, impacted by lifestyle choices and aging mechanisms, results in significant physiological changes with potential health implications [14].

Further investigations in our study uncovered the relationship between CRC stage and BMI and showed that patients with higher BMIs have a higher prevalence of CRC in stage 3 (*p* = 0.02) and patients with lower BMIs have a higher prevalence of CRC stage 4 (*p* = 0.002). However, a study that partially aligns with our study conducted by Spychalski et al. found a negative trend between stage and BMI group (*p* = 0.014), meaning that with a higher BMI, the stage of CRC was lower [8]. This difference in findings might be due to differences in the study population, methodology, sample size, or weight loss associated with cancer. We believe that the weight loss observed in Stage 4 CRC patients can be attributed to a complex interplay of various factors within the tumor microenvironment (TME) and the tumor macroenvironment. A literature review highlights the significant role of pro-inflammatory cytokines, such as TNF-α, IL-6, and IL-1, and certain chemokines, like CXCL8/IL-8, as procachectic agents produced by different TME cell types in CRC-associated cachexia. However, assessing the role of single TME cells in CRC-associated cachexia remains a challenging task due to the lack of an ideal CRC-induced in vivo cachexia model, overlapping mechanisms and signaling pathways in CRC progression, and the interrelated nature of cancer-induced cachexia. Further research is necessary to better understand the detailed mechanisms of malnutrition in CRC and the role of autophagy processes associated with TME cells in inducing human cancer cachexia [15].

### 4.1. Risk Factors Associated with the Detection of Adenomas and CRC

Kobiela et al. 2018 investigated the relationship between BMI and CRC screening outcomes in 75,278 participants grouped by BMI and studied the adenoma detection rate (ADR), advanced neoplasia detection rate (ANDR), and cancer detection rate. Adenomas are benign tumors that can eventually progress to CRC. The researchers found that both the ADR and ANDR increased significantly with increasing BMI in both proximal and distal locations. Although the number of cancers detected increased with higher BMIs, statistical significance was not reached. There was, however, a trend towards a higher incidence of proximal cancers in groups with increasing BMIs (*p* = 0.065). To give more statistical analysis power, the multivariate logistic regression models showed that an increasing BMI was associated with an increased risk of detecting adenomas and advanced neoplasia, with stronger associations in proximal locations [16]. While the study mentioned above did not achieve statistical significance in the number of cancers detected in the group with higher BMI, its findings support the results of our own research, where we observed a higher incidence of cancer in the distal part of the colon. Additionally, in the age group of elderly patients, there was a nearly significant result for CRC in proximal locations, with an odds ratio of 1.69 [0.98; 2.93] and a *p*-value of 0.0589.

Other factors that can contribute to an increased risk of adenomas are reported in the literature. A study found a significantly higher mean age in the adenoma-positive group compared to those with normal colonoscopy results (*p* < 0.001). Furthermore, the presence of a positive history of type 1 or type 2 diabetes was also found to be significantly more prevalent in the adenoma-positive group. In addition, the incidence of a positive family history of colorectal cancer was notably higher among adenoma-positive patients as compared to those with normal colonoscopy outcomes. These findings highlight the importance of age, type 1 and 2 diabetes, and family history as potential risk factors for the development of adenomas [17].

### 4.2. Loss of SIRT6 and Adipose Tissue Homeostasis

In a study by Sebastian et al. 2022, where mouse models were utilized, specifically APCmin mice, it was determined that the loss of SIRT6 resulted in an increase in the number and size of intestinal adenomas. They discovered that SIRT6 influences tumor-initiating cells (TICs), which are responsible for tumor formation and growth. The loss of SIRT6 led to an expansion of the intestinal stem cell (ISC) pool, and this increase was associated with changes in glucose metabolism. Treatment with an inhibitor of glycolysis, dichloroacetate (DCA), reversed these effects, indicating that increased glucose metabolism upon SIRT6 loss drives ISC expansion, leading to an increase in TICs [18].

To create an overview of the functions of SIRT6 protein in adipose tissue, we found a literature review that explains the physiological and pathological pathways of the protein. In summary, Sirt6 plays a crucial role in adipocyte metabolism, inflammation, and immune cell function. Its main functions in adipocytes include regulating lipid metabolism, preventing inflammation, stimulating lipolysis, enhancing adipose tissue browning, and improving insulin action in peripheral tissues. Consuming a high-fat diet leads to a notable decrease in Sirt6 levels within visceral adipose tissue, which is then followed by an increase in body weight, fat mass, inflammation within the adipose tissue, and insulin resistance. Sirt6 is also involved in macrophage polarization, influencing both M1 and M2 types. In M1-type macrophages, Sirt6 deficiency activates the NF-κB pathway, producing IL-6 and activating STAT3, leading to M1-type macrophage polarization. In M2-type macrophages, Sirt6 activates the PI3K-Akt pathway and increases the expression of M2 marker genes [19].

Building on this foundation of understanding, a recent study by Jae Hoon Lee and colleagues provides further insights into the significance of adipose tissue characteristics in relation to health outcomes, especially in the context of CRC. In this study, various factors were assessed for their potential association with overall survival (OS). The study utilized a multivariate model incorporating the volume ratio of visceral adipose tissue (rVAT) and the standardized uptake value of visceral adipose tissue (VAT-SUV) in both male and female populations. When applying the multivariate model with rVAT and VAT-SUV to both genders, advanced cancer stages and chemotherapy remained significantly associated with OS across both sexes. Moreover, the presence of lymphovascular invasion and a high–high combination of rVAT and VAT-SUV were additionally significant for males. For females, any combination involving a high component of rVAT and/or VAT-SUV increased the risk of a poor OS [20].

Overall, overweight and obesity are marked by disruptions in glucose metabolism such as insulin resistance and hyperglycemia, and could potentially be exacerbated by the absence of Sirt6 that leads to an expansion of ISCs and TICs, thereby increasing the risk of adenoma development and colorectal cancer.

These findings align with our understanding of the role of Sirt6, emphasizing the need for further research into the intricate interactions between Sirt6 functions and the volume and metabolic activity of visceral adipose tissue in male and females, potentially leading to more effective interventions for adverse health outcomes related to cancer.

### 4.3. Inflammation in the Colorectal Mucosa

A study by Mariani et al. 2017 investigated the relationship between inflammation in the colorectal mucosa, obesity, and adenomas. Inflammation was assessed using myeloperoxidase (MPO) positive cells in the mucosa. A significant increase in MPO-positive cells was found in obese subjects compared to non-obese subjects. Overweight and obese subjects also showed a significantly higher number of MPO-positive cells in their colorectal mucosa. During follow-up, adenoma occurrence was significantly higher in subjects with a higher grade of inflammation in their colorectal mucosa [21].

MPO is a heme-containing enzyme primarily found in neutrophils and monocytes, which plays a crucial role in the body’s innate immune system. It is involved in various physiological processes, such as the killing of microorganisms, recruitment of polymorphonuclear leukocytes (PMNs) to inflamed tissues, regulation of apoptosis, modulation of PMN–macrophage interactions, formation of lipophilic chloramines, and binding to serum proteins. These diverse functions allow MPO to contribute to host defense, regulate inflammation, and maintain overall immune system homeostasis [22].

In our effort to uncover more correlations with inflammation in colorectal mucosa, we discovered studies in the literature that discuss the diversity of the microbiome in patients with adenoma or colorectal cancer and its potential link to inflammation.

Wu et al.’s 2021 research uncovers distinct variations in microbial communities during CRC progression. They found a considerable variation in differential amplicon sequence variants (ASVs) and markers for differentiating adenoma and cancer from healthy controls. Among these, *E. ruminantium* was identified as a common adenoma-associated marker, while several other bacteria, including *Porphyromonas* sp. *HMSC077F02, L. pectinoschiza*, and *Hungatella hathewayi WAL-18680*, were recognized as common cancer-associated biomarkers. Notably, *F. nucleatum*, a prevalent CRC biomarker, was not identified as a differential bacterium or biomarker between controls and adenomas. The study further demonstrated that coupling adenoma-specific markers with the fecal immunochemical test (FIT) enhanced the early adenoma detection accuracy, suggesting a complementary role for the non-invasive FIT test alongside gut microbiota analyses [23].

The study also pinpointed the biosynthesis of ADP-heptose and the key gene hldE as significantly enriched in adenoma versus controls. The amplified activity of the ADP-heptose biosynthesis pathway, a known bacteria-related carcinogen, may trigger prolonged aggravation of NF-κB signaling during CRC evolution. Intriguingly, the abundance of ADP-heptose biosynthesis was substantially increased in adenoma but not in CRC compared to adenoma, indicating a pivotal role for ADP-heptose in adenoma. In addition, the team identified substantial differences in vitamin K2 biosynthesis genes, such as menH and menF, between adenoma and cancer. Vitamin K2, known for its antitumor effects via cell cycle arrest, differentiation, and apoptosis, might see increased production as a compensatory effect of dysregulated microbiota within the tumor environment, hinting at a potential CRC intervention strategy targeting vitamin K2 biosynthesis bacteria [23].

In another study, the researchers found that *Parvimonas micra (P. micra),* a bacterium, is significantly enriched in both fecal samples and tissue biopsies of CRC patients compared to healthy subjects. *P. micra* was also identified as an independent predictor of a poor outcome in CRC patients. Through various experiments on mice and in vitro using human colon adenocarcinoma cell line HT-29, the researchers demonstrated that *P. micra* promotes colorectal tumorigenesis, colonocyte proliferation, and activates the Wnt/β-catenin signaling pathway. Additionally, the study showed that *P. micra* alters the expression of cell-proliferation-related genes, induces Th17 immune cell infiltration, and promotes the establishment of a pro-inflammatory microenvironment, which favors colorectal tumorigenesis. Overall, these results suggest that *P. micra* could be a predictor of a poor prognosis in patients with CRC and that it plays a role in promoting colorectal cancer development [24].

Furthermore, a study by Kneis et al. found that the tumor tissue of right-sided colon cancer (RSCC) patients was characterized by a significant increase in the abundances of *Haemophilus* and *Veilonella*, while increased abundances of *Bifidobacterium, Akkermansia, Roseburia*, and *Ruminococcus* were associated with left-sided colon cancer (LSCC). Moreover, grade 3 tumors were significantly enriched in *Fusobacterium* and *Parvimonas*, while grade 2 tumors were significantly enriched in *Fusicatenibacter, Blautia, Intestimonas*, and *Romboutsia* [25].

Summarizing the latest research on the inflammation of colorectal mucosa, it appears that obesity is most likely linked to the prevalence of MPO-positive cells within the colorectal mucosa, suggesting an underlying inflammatory response. Complementary to this, studies have shed light on the pivotal role that variations in microbial communities and functional pathways play in the onset of CRC. Particularly noteworthy are the adenoma-specific signatures which hold promise for early detection and pave the way for early interventions. Additionally, it has come to light that RSCC and LSCC, due to their distinct microbiome characteristics, should be studied further, each warranting a bespoke screening or therapeutic approach. Adding another layer of complexity, the bacterium *P. micra* has emerged as a potential sign of a poor CRC prognosis, hinting at its involvement in the disease’s progression.

### 4.4. Systemic Gene Expression and Visceral Obesity

Cariello et al. 2022 explored the role of microRNAs (miRNAs) in platelets, small RNA molecules that regulate gene expression and can be transferred between cells. The researchers found a different miRNA profile in the platelets of visceral obesity (VO) patients compared to healthy controls, with certain miRNAs, like miR-19a-5p, significantly upregulated. Notably, miR-19a-5p was also found to be elevated in the plasma of VO patients, suggesting a systemic upregulation [26].

The researchers also examined miR-19a expression in human colon cancer samples and discovered an increase in miR-19a expression in colon cancer and adenoma samples compared to normal mucosa. They found that miR-19a was implicated in several cancer-related signaling pathways. The study also found that direct injection of miR-19a into a xenograft tumor model led to accelerated tumor growth and increased proliferating cell nuclear antigen accumulation. This suggests that the unique miRNA profile in the platelets of VO patients could play a part in promoting tumor growth [26].

## 5. Conclusions

Our study presents compelling evidence that obesity significantly elevates the risk of CRC, with obese patients demonstrating a markedly higher prevalence of rectal cancer compared to their normal weight counterparts. This is further highlighted by our observation of a relationship between CRC stage and BMI, where patients with higher BMIs were more likely to present with CRC in stage 3, while those with lower BMIs were more predisposed to stage 4 CRC.

Interestingly, our investigation also revealed distinct differences across age groups in the normal weight and overweight categories, with a notable trend of increasing mean weight values with age. However, this pattern was not observed in the obese category, indicating that the relationship between age and weight may vary depending on the BMI range. Within the elderly patient group, we detected a borderline significant association between CRC in proximal locations and age. While this result did not reach statistical significance, it suggests a potential trend that requires further investigation.

In addition to the aforementioned findings, the body of research we have surveyed provides compelling evidence for the multifactorial and complex nature of colorectal cancer pathogenesis, which is intricately tied to factors such as adipose tissue homeostasis, inflammation in the colorectal mucosa, gut microbiota diversity, and systemic gene expression. The critical role of SIRT6 in maintaining adipose tissue homeostasis, disclosed by the loss of SIRT6 that has been found to increase the number and size of intestinal adenomas, suggests the importance of this protein in preventing CRC initiation and progression.

Simultaneously, adipose tissue characteristics and their relationship with CRC prognosis have been highlighted, particularly in relation to sex-specific metabolic activity and volume. The link between obesity, indicated by increased visceral adipose tissue and systemic inflammation, and CRC further highlights the need for a multidimensional understanding of CRC pathogenesis.

The role of inflammation in the colorectal mucosa, as indicated by the presence of MPO-positive cells, and its association with obesity and adenoma occurrence is another critical aspect to consider. This is further complicated by the emerging evidence that distinct variations in microbial communities can impact CRC progression. The identification of specific bacterial markers associated with adenoma and CRC, as well as the potential role of certain bacteria in promoting CRC, opens up promising paths for early detection and intervention strategies.

Lastly, systemic gene expression, particularly the role of miRNAs, adds another dimension to our understanding of CRC. The unique miRNA profile in the platelets of visceral obesity patients and the potential role of these small RNA molecules in promoting tumor growth indicate the need for further investigation into the mechanisms of CRC at the molecular level.

In conclusion, these findings collectively paint a picture of CRC as a disease influenced by a multitude of interconnected factors. They underscore the need for a holistic, integrative approach to its study and treatment, considering not only individual genetic and physiological factors but also the complex interplay between these elements. Further research is needed to fully understand these interactions and their implications, with the ultimate goal of developing more effective, personalized interventions to prevent and treat CRC.

### Limitations

This study has several limitations that should be acknowledged. Firstly, the sample size of 830 patients, while substantial, may not be large enough to detect smaller effects or associations, particularly for less common categories of cancer locations. Furthermore, the study design was cross-sectional, limiting the conclusions that can be drawn from the results to associations, not causality.

Secondly, the study might not have adequately controlled all potential confounding factors. Although the analysis considered age, sex, BMI, and cancer stage, other important factors such as dietary habits, physical activity, smoking status, alcohol consumption, family history of cancer, or other comorbidities were not accounted for in the model. These unmeasured confounders could potentially influence the association between cancer location and the demographic and health variables of interest.

This study relied on existing medical records for data collection, which may have inherent inaccuracies or inconsistencies. Moreover, the availability and quality of the data could have been limited by what was recorded in these records.

Furthermore, the study’s reliance on BMI as a key variable is a potential limitation. BMI, while widely used, does not distinguish between fat and muscle mass and might not fully capture an individual’s health status or risk profile.

This study also focused mainly on rectal and ascending colon cancers. The findings may not be applicable to other locations of colorectal cancer or more generally to other types of cancer.

Lastly, this study did not incorporate longitudinal follow-up, which could provide additional insights into disease process over time, including changes in the patient’s health status or cancer progression.

Future studies should aim to address these limitations by incorporating a more diverse sample considering additional confounding factors (such as the waist/hip ratio, volume and metabolic activity of visceral adipose tissue, and others) and potentially using a longitudinal design to track disease progression over time.

## Figures and Tables

**Figure 1 medicina-59-01399-f001:**
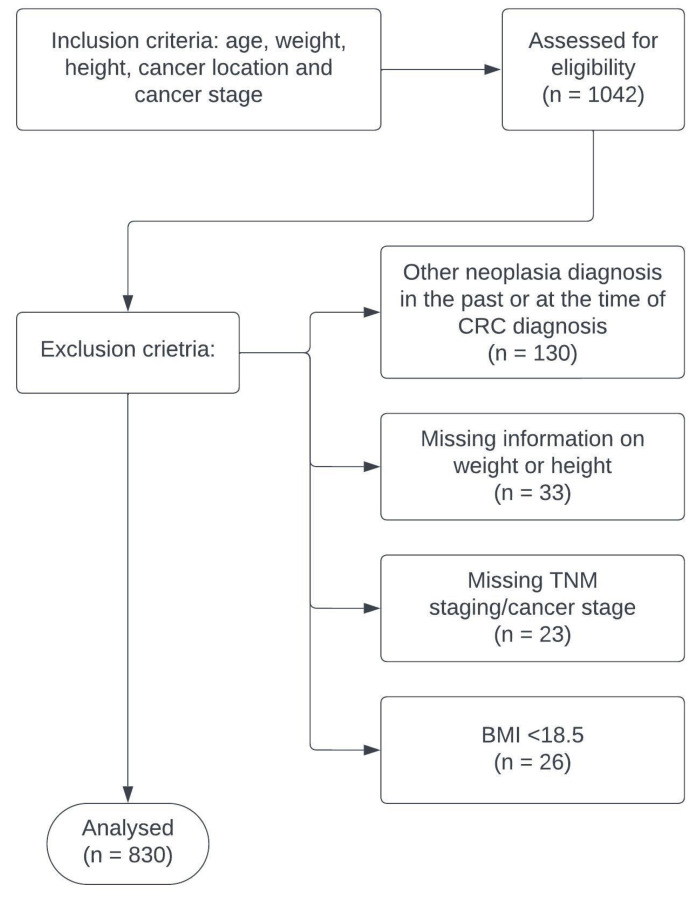
Consort flow diagram for inclusion and the reasons for exclusion of all the samples.

**Figure 2 medicina-59-01399-f002:**
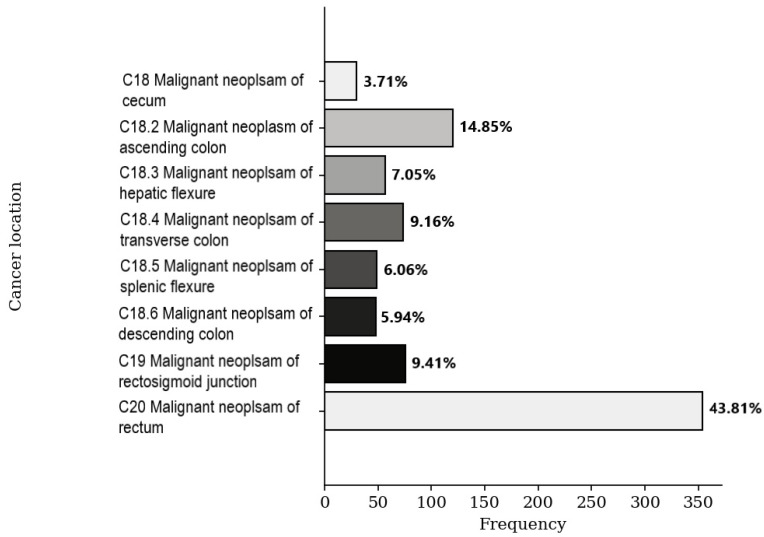
Frequency distribution of colorectal cancer.

**Figure 3 medicina-59-01399-f003:**
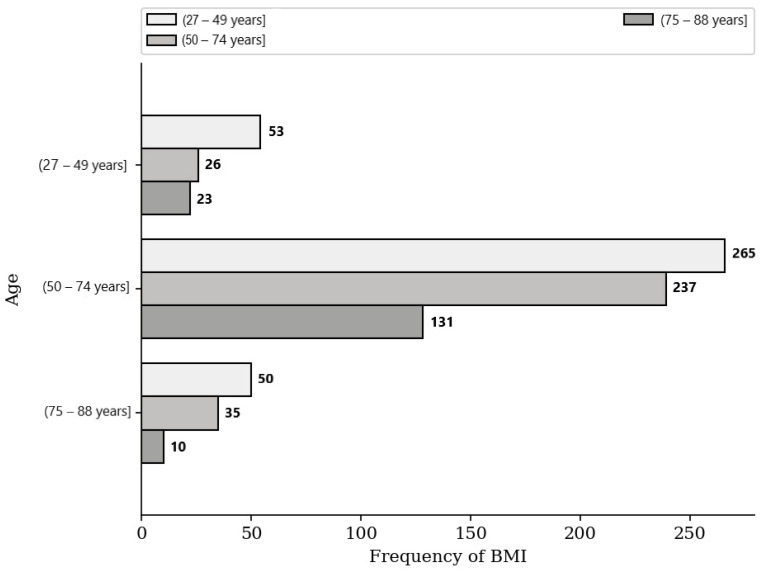
Frequency of BMI based on age subgroups.

**Table 1 medicina-59-01399-t001:** Distribution of colorectal cancer subtypes across different age groups.

Variable	Young Adults (27–49 Years)	Middle-Aged Adults (50–74 Years)	Elderly Adults (75–88 Years)	Total
C18 Malignant neoplasm of cecum	3	21	6	30 3.71% (N = 808)
	10.0%	70.0%	20.0%	
	3.03% (N = 99)	3.41% (N = 616)	6.45% (N = 93)	
	0.37% (N = 808)	2.6% (N = 808)	0.74% (N = 808)	
C18.2 Malignant neoplasm of ascending colon	15	85	20	120 14.85%
	12.5%	70.83%	16.67%	
	15.15%	13.8%	21.51%	
	1.86%	10.52%	2.48%	
C18.3 Malignant neoplasm of hepatic flexure	4	48	5	57 7.05%
	7.02%	84.21%	8.77%	
	4.04%	7.79%	5.38%	
	0.5%	5.94%	0.62%	
C18.4 Malignant neoplasm of transverse colon	10	57	7	74 9.16%
	13.51%	77.03%	9.46%	
	10.1%	9.25%	7.53%	
	1.24%	7.05%	0.87%	
C18.5 Malignant neoplasm of splenic flexure	5	38	6	49 6.06%
	10.2%	77.55%	12.24%	
	5.05%	6.17%	6.45%	
	0.62%	4.7%	0.74%	
C18.6 Malignant neoplasm of descending colon	4	42	2	48 5.94%
	8.33%	87.5%	4.17%	
	4.04%	6.82%	2.15%	
	0.5%	5.2%	0.25%	
C19 Malignant neoplasm of rectosigmoid junction	10	60	6	76 9.41%
	13.16%	78.95%	7.89%	
	10.1%	9.74%	6.45%	
	1.24%	7.43%	0.74%	
C20 Malignant neoplasm of rectum	48	265	41	354 43.81%
	13.56%	74.86%	11.58%	
	48.48%	43.02%	44.09%	
	5.94%	32.8%	5.07%	
Total	99	616	93	808
	12.25%	76.24%	11.51%	100%

**Table 2 medicina-59-01399-t002:** Comparison of body mass index across different age groups.

Variable	Young Adults (27–49 Years) N = 102	Middle-Aged Adults (50–74 Years) N = 633	Elderly Adults (75–88 Years) N = 95	*p*-Value
NW	21.96 (±1.49)	22.47 (±1.75)	22.74 (±1.73)	0.022
	Range: (19.1; 24.5)	Range: (18.7; 24.9)	Range: (18.7; 24.9)	
	N = 53	N = 265	N = 50	
OW	26.52 (±1.26)	27.31 (±1.35)	27.26 (±1.26)	0.015
	Range: (25.0; 29.4)	Range: (25.0; 29.9)	Range: (25.1; 29.7)	
	N = 26	N = 237	N = 35	
OB	34.23 (±3.17)	33.44 (±3.44)	32.26 (±2.13)	0.16
	Range: (30.1; 40.9)	Range: (30.0; 46.6)	Range: (30.1; 35.9)	
	N = 23	N = 131	N = 10	

**Table 3 medicina-59-01399-t003:** Comparison of cancer stages across different body mass index groups.

Variable	NW (18.5–24.9) N = 370	OW (25.0–29.9) N = 300	OB (30.0–47.0) N = 160	*p*-Value
Stage 1				0.335
Yes	11 (2.97%)	11 (3.67%)	2 (1.25%)	
No	359 (97.03%)	289 (96.33%)	158 (98.75%)	
	N = 370	N = 300	N = 160	
Stage 2				0.43
Yes	88 (23.78%)	82 (27.33%)	36 (22.5%)	
No	282 (76.22%)	218 (72.67%)	124 (77.5%)	
	N = 370	N = 300	N = 160	
Stage 3				0.02
Yes	134 (36.22%)	134 (44.67%)	76 (47.5%)	
No	236 (63.78%)	166 (55.33%)	84 (52.5%)	
	N = 370	N = 300	N = 160	
Stage 4				0.002
Yes	137 (37.03%)	73 (24.33%)	46 (28.75%)	
No	233 (62.97%)	227 (75.67%)	114 (71.25%)	
	N = 370	N = 300	N = 160	

**Table 4 medicina-59-01399-t004:** Odds ratios for malignant neoplasm of the rectum and ascending colon based on age, body mass index, cancer stage, and gender.

		Malignant Neoplasm of the Rectum		Malignant Neoplasm of Ascending Colon	
Variable	Modality	Odds Ratio	*p*-Value	Odds Ratio	*p*-Value
Intercept		0.713 [0.543; 0.938]	0.0157	0.13 [0.0874; 0.195]	1.87 × 10^−23^
Age	(50–74 years) (reference)				
	(27–50 years)	1.24 [0.813; 1.9]	0.315	1.13 [0.622; 2.05]	0.69
	(74–88 years)	1.12 [0.721; 1.74]	0.613	1.69 [0.98; 2.93]	0.0589
BMI	NW (18.5–24.9) (reference)				
	OW (25.0–29.9)	1.16 [0.848; 1.59]	0.353	1.15 [0.747; 1.77]	0.529
	OB (30.0–47.0)	1.56 [1.07; 2.27]	0.0213	0.882 [0.506; 1.54]	0.657
Cancer stage		0.771 [0.563; 1.06]	0.105	1.1 [0.715; 1.68]	0.674
Gender		0.864 [0.653; 1.14]	0.305	1.27 [0.861; 1.88]	0.226

## Data Availability

The data generated or analyzed during this study are included in this published article or are available from the corresponding author on reasonable request.

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
