# Peer review of "Exploring the Influence of Age, Gender and Body Mass Index on Colorectal Cancer Location"

_medicina, 2023, doi:10.3390/medicina59081399_

Round 1

Reviewer 1 Report

Hello

Your research is relevant and well done, but there are a few questions that clarify the content

1. How was the sample of patients (1043 people) determined? Only according to the criteria that you derived or was a preliminary calculation of the sample size determined?

2. Did you receive permission from the ethical committee for the study?

3. In the results, I would suggest using graphs-diagrams for clarity, and not just tables of the epidemiology of the disease

4. Figure 2 should be corrected and made more representative

5. According to the list of references, I have no questions

Author Response

1. The ethics commission approved the study for the period from January 1, 2016, to March 1, 2023.

2. Yes, the study was approved by the ethics committee. I have updated the 'Materials and Methods' section (pages 99-100).

3. We believe that for this article, the raw data (tables) provide a more accurate representation of the information. We have also added a graphical diagram (pages 194-195).

4. Figure 2 has been corrected and made more representative.

Reviewer 2 Report

The authors had a study on he influence of age, gender and body mass index on colorectal cancer location.

The subject is not novel. however, it is a main concern in its field. The authors can explain novelty.

My comments:

The conclusion section of the abstract should be improved. It does not support the main results of the study. 

The introduction does not provide sufficient background and does not include all relevant references. It is also short. It must be improved and some recent and relevant references should be added.

The cited reference numbers are not according to the journal format. The points should be inserted after numbers.

What was the base and the reason for data Extraction? Please more detail.

Please improve the quality of Figure 1.

"Twenty-two patients were not included in this table due to unspecified colon cancer location" . Why? explain.

Please add some figures.

The conclusion is so short and does not support the main results. Please improve.

Minor editing of English language required

Author Response

1. The conclusion section of the abstract has been improved by reintroducing the main results of the study (lines 34-43).

2. While I appreciate the feedback about the introduction, I would argue that its current length and content are both appropriate and adhere to the guidelines stipulated by MDPI.

 - As stated on the MDPI's website, there is no explicit word count requirement for individual sections of the manuscript, including the introduction. The minimum word count requirement of 4000 words applies to the entire article and doesn't dictate the distribution of those words across different sections. Therefore, it would be inaccurate to deem the introduction insufficient based on its length.

 - The emphasis should be on the quality of content rather than the quantity of words. A clear and concise introduction that successfully outlines the research question and provides the necessary context is more valuable than a lengthy introduction that dilutes the main points or overwhelms the reader.

 - All references used in the introduction are from 2023, signifying that the background provided is based on the most recent and relevant research.

 - Based on the feedback, additional recent studies (line 71-86) have been incorporated to provide further background. This adjustment ensures a more comprehensive understanding of the context without unnecessarily extending the length of the introduction.

 - Considering that MDPI publications are aimed at a specialist audience, the introduction does not need to include extensive basic background information, as this could be considered redundant to such readers.

3. The cited reference numbers have been updated as requested.

4. The qulity of Figure 1 has been improved.

5. This study analyzed the subgroups of colorectal cancer. Including the twenty-two patients who don't contribute meaningful or specific information to any of the defined subgroups would not be beneficial.

6. A figure has been added as requested (lines 194-195).

Round 2

Reviewer 2 Report

It can be accepted.